# A Clinicopathological Analysis of Asian Patients with Adrenocortical Carcinoma: A Single-Center Experience

Wen-Hsuan Tsai [1] , Shuen-Han Dai [2], Chun-Chuan Lee [1,3], Ming-Nan Chien [1,3] and Yi-Hong Zeng [1,3,*]

1   Division of Endocrinology and Metabolism, Department of Internal Medicine, MacKay Memorial Hospital, Taipei 104, Taiwan; u9701003@cmu.edu.tw (W.-H.T.)
2   Department of Pathology, MacKay Memorial Hospital, Taipei 104, Taiwan
3   Department of Medicine, MacKay Medical College, New Taipei City 252, Taiwan
*   Correspondence: starrydouchain@gmail.com; Tel.: +886-2-2543-3535 (ext. 2173); Fax: +886-2-2523-2448

**Abstract:** Background: There is limited information regarding the immunohistochemistry stain and its prognostic role in adrenocortical carcinoma (ACC), and few studies focus on Asian patients. Our study aims to identify the correlation between immunohistochemistry staining and the prognosis of ACC in Asian patients. Methods: We searched the database of a single center in Taiwan for cases with a pathological diagnosis of ACC in the past 25 years. We collected patient data on age, sex, initial presentation, staging, metastatic site, and survival duration. Immunohistochemical studies using antibodies to CDK4, ATRX, beta-catenin, Ki-67, SSTR2, and p53 were performed. Survival analysis was performed using the log-rank test, the Cox proportional hazards model and bootstrapping with 5000 samplings. Results: Fourteen patients were identified, and the median age was 49.5 (range 1–70) years. There were eight male and six female patients. Four patients presented with Cushing's syndrome, and half were diagnosed with stage IV ACC at presentation. Only three patients survived (21%). The median survival time was 15.5 (range 0.67–244) months. SSTR2 expression score > 50 (log-rank test: $p = 0.009$) and Ki-67 > 50% (log-rank test: $p = 0.017$) were associated with mortality. However, after adjusting for stage, the bootstrapping analysis demonstrated that Ki-67 [B 2.04, $p = 0.004$], Beta-catenin [B 2.19, $p = 0.009$], ATRX [B 1.48, $p = 0.026$], P53 [B 1.58, $p = 0.027$], SSTR2 [B 1.58, $p = 0.015$] and SSTR2 expression score [B 0.03, $p < 0.001$] were all significantly associated with mortality. Conclusions: After adjusting for stage, Ki-67 > 50%, Beta-catenin, ATRX, P53, SSTR2 and SSTR2 expression score > 50 were associated with mortality in Asian patients with ACC.

**Keywords:** adrenocortical carcinoma; immunohistochemistry stain; SSTR; Ki-67





## 1. Introduction

Adrenocortical carcinoma (ACC) is rare and has a poor prognosis. Even for patients with resected tumors, the median survival is only 32 months in the United States [1]. ACC can occur at any age, with a peak incidence between 40 and 50 years, and has a female predominance (55–60%) [2,3]. At least 50–60% of those with ACC show clinical hormone excess, with Cushing's syndrome being the most common [4]. The 5-year survival of ACC is 60–80% for tumors confined to the adrenal space, 35–50% for locally advanced disease, and much lower in metastatic disease with 0–28% [4]. Information on the incidence and prognostic factors of ACC in Asian patients is limited. The Korean Endocrine Society conducted a registry-based nationwide survey and reported that the median age at diagnosis of ACC was 51.5 years, with a female predominance ($n = 110$, 53.9%). Abdominal pain was the most frequent symptom ($n = 70$, 40.2%), and stage II was the most common ($n = 62$, 30.4%) at the time of diagnosis [5]. Hypercortisolism and elevated Dehydroepiandrosterone sulfate (DHEAS) were the most common manifestations of a functional tumor [5]. The remission rate was 48%, and the median recurrence-free survival time was 46 months. The estimated 5-year overall survival (OS) and disease-specific survival rates were 64.5% and 70.6%,

respectively [5]. Higher staging, larger tumor size, and the presence of a cortisol-secreting tumor were risk factors for ACC-specific death [5].

The poor prognosis of ACC is associated with tumor grade, mitotic count, Ki-67 proliferation index [6], resection status [1], and results of p53 and CTNNB1 (beta-catenin) immunohistochemistry (IHC) [7]. However, the prognostic role of the somatostatin receptor (SSTR) remains unestablished. Somatostatin analogs have shown promising effects in the treatment of multiple neuroendocrine tumors [8]. However, very little information is available regarding the effect of somatostatin analogs on ACC. Our aim is to present the clinical and pathological pattern of ACC in Asian patients and to evaluate IHC expression and its prognostic role in ACC.

## 2. Methods

### 2.1. Subjects

We collected paraffin-embedded archival pathologic specimens from 14 patients with ACC from the archives of the Department of Pathology, MacKay Memorial Hospital, between July 1999 and January 2020. All of the tissues were histologically reviewed, and the representative sections from the formalin-fixed paraffin-embedded block of the 14 cases were selected. The Mackay Memorial Hospital Institutional Review Board approved the study protocol (20MMHIS473e).

### 2.2. Clinicopathologic Information

The patient's clinical information was obtained from their medical records in the MacKay Memorial Hospital. The records were reviewed for patient age, sex, initial presentation, staging, metastatic site, treatment, survival duration, hormone profile, mitotic count, Fuhrman nuclear grade, resection margin and tumor rupture status. Hematoxylin and eosin–stained slides and IHC studies were reviewed when available.

### 2.3. Immunohistochemistry

The IHC analysis for beta-catenin (Bio-Genex Laboratories, Fremont, CA, USA, cat# NU510-UC), p53 (Cell Marque Corporation, Rocklin, CA, USA, cat# 453M-96), ATRX (Bio SB, Inc., Santa Barbara, CA, USA, cat# BSB 3297), CDK4 (Bio SB, Inc., Santa Barbara, CA, USA, cat# BSB 2465), and SSTR2 (Santa Cruz Biotechnology, Santa Cruz, CA, USA, cat#sc-365502) was performed. Nuclear expression of beta-catenin and CDK4, complete loss of ATRX, and aberrant expression of p53 (complete loss or diffuse strong expression) were recorded. SSTR2 staining intensity was scored as 0 (no staining), 1+ (weak staining), 2+ (moderate staining), and 3+ (strong staining), and expression scores were calculated by multiplying the intensity and percentage of positively stained cells (0–100%). The maximal score is 300. The Ki-67 index was also recorded and classified with a cut-off point of $\leq 50\%$ and $>50\%$.

### 2.4. Statistical Analysis

Survival analysis was performed using the log-rank test. Kaplan-Meier survival curves were generated, and differences between the curves were examined by log-rank testing. Cox proportional hazards model with a hazard ratio (HR) and 95% confidence interval (CI) and a structural equation modeling framework with 5000 bootstrapping samplings were conducted. The stage of ACC was further adjusted. The time from diagnosis to death resulting from any cause was defined as the OS. Censored observations were defined as patients alive at the date of the last follow-up or lost follow-up (31 October 2020). All tests were two-sided, and $p < 0.05$ was considered statistically significant. IBM SPSS Statistics 28.0 was used for statistical analysis.

## 3. Results

Fourteen patients were identified, and the median age was 49.5 (range 1–70) years. There were eight male and six female patients. Four patients presented with Cushing's

syndrome, while half of the patients were diagnosed with stage IV ACC at presentation (Table 1). The most frequent metastatic site was in the liver, followed by the lung and peritoneum. Only three patients survived (21%); the median survival time was 15.5 (range 0.67–244) months. Three of the patients suffered from second malignancy, including testis cancer, endometrial mullerian adenosarcoma, and hepatocellular carcinoma. The results of IHC staining for 14 patients are summarized in Table 2. Of the 14 patients, eight (57%) had positive CDK4 nuclear expression, seven (50%) had positive ATRX nuclear expression, and only one patient (7%) had positive beta-catenin nuclear stains. Ten of the 14 patients (71%) had positive SSTR2 stains, and the median SSTR2 expression score was 83 (range 0–270). Three of the patients (21%) had diffuse positive staining of p53, while eight patients (57%) had total negative staining of p53.

**Table 1.** Clinical presentation of 14 patients with ACC.

| No. | Age | Sex | Initial Presentation | Tumor Size (cm) and Laterality | Stage | Metastatic Site | Treatment | Survival Duration (Months) | Survival Status | Hormone Profile | Other Malignancy |
|---|---|---|---|---|---|---|---|---|---|---|---|
| 1 | 5 | M | penis enlargement, virilization, and acne | 7.4, unilateral | 4 | liver, retroperitoneum | Adrenalectomy and mitotane | 15 | expired | elevated DHEAS | |
| 2 | 1 | M | penis enlargement, virilization | 12, unilateral | 4 | peritoneum seeding | Adrenalectomy | 14 | expired | | |
| 3 | 70 | M | retroperitoneal tumor | 5, unilateral | 1 | NA | Adrenalectomy | 61 | expired | | testis cancer |
| 4 | 44 | M | Severe abdominal pain for 2 days | 9.5, unilateral | 3 | NA | Adrenalectomy and mitotane | 21 | expired | | |
| 5 | 60 | M | body weight loss about 11 kg in recent 5–6 months | 11, unilateral | 4 | lung | Adrenalectomy, mitotane, chemotherapy with DEP regimen | 144 | alive | | |
| 6 | 1 | F | progressive rapid weight gain for months, hirsutism, pubic hair | NA | 3 | NA | Adrenalectomy | 0.67 | expired | | |
| 7 | 62 | M | enlarged soft tissue mass in left adrenal gland | 11.5, unilateral | 4 | liver, kidney, pancreas, diaphragm, small and large intestine | Adrenalectomy, chemotherapy with doxorubicin | 16 | expired | | HCC |
| 8 | 49 | F | general weakness, nausea, and dizziness for 2 weeks, acne, right adrenal tumor | 7, unilateral | 4 | suspected bone metastasis | Adrenalectomy | 90 | expired | Cushing syndrome, parathyroid adenoma, pituitary tumor, elevated DHEAS | Endometrial mullerian adenosarcoma |
| 9 | 50 | M | progressive abdomen distension for months | 13.1, unilateral | 4 | suspected liver and lung | No treatment | 0.4 | expired | Cushing syndrome, elevated estrogen/ DHEAS/ 17OHP | |
| 10 | 40 | F | amenorrhea at 40 y/o | 11, unilateral | 3 | NA | Adrenalectomy, chemotherapy with unknown regimen | 244 | expired | Cushing syndrome | |
| 11 | 66 | F | suprarenal mass noted via renal echo | 9.5, unilateral | 2 | local recurrence, retroperitoneum | Adrenalectomy | 5 | expired | | |

**Table 1.** *Cont.*

| No. | Age | Sex | Initial Presentation | Tumor Size (cm) and Laterality | Stage | Metastatic Site | Treatment | Survival Duration (Months) | Survival Status | Hormone Profile | Other Malignancy |
|---|---|---|---|---|---|---|---|---|---|---|---|
| 12 | 36 | F | abdomen pain, body weight increase, acne, buffalo hump | 7.7, unilateral | 3 | lung | Adrenalectomy, mitotane, radiotherapy, metastasis resection, chemotherapy with DEP regimen | 24 | alive | Cushing syndrome, elevated testosterone/ DHEAS/ ASD/ 17OHP | |
| 13 | 56 | M | abdomen pain and fullness for 2 years | 9.2, unilateral | 2 | NA | Adrenalectomy and mitotane | 15 | alive | | |
| 14 | 59 | F | abdomen pain and fullness | 23, unilateral ovary (ectopic ACC) | 4 | lung, liver, peritoneum | Laparotomy optimal cytoreduction, mitotane and chemotherapy with etoposide+ cisplatin | 6 | expired | | |

DEP: doxorubicin, etoposide, cisplatin.

**Table 2.** Results of immunohistochemical staining and pathological characteristics for 14 patients with ACC.

| No. | Beta-Catenin (Nuclear Stain) | CDK4 (Nuclear Expression) | ATRX (Nuclear Expression) | p53 (Aberrant Expression) | SSTR2 | SSTR2 Expression Score | Ki-67 | Mitotic Count | Fuhrman Nuclear Grade | Resection Margin Status and Tumor Rupture |
|---|---|---|---|---|---|---|---|---|---|---|
| 1 | − | + | + | Diffuse (+) | + | 100 | 10 | NA | 3 | R0, no rupture |
| 2 | − | + | + | Diffuse (+) | + | 155 | 5 | >31/50HPF | 4 | R0, no rupture |
| 3 | − | + | + | Total (−) | − | 0 | <1 | NA | 4 | R0, no rupture |
| 4 | − | + | − | Mosaic | + | 70 | >4 | >5/50 HPF | 2 | R0, no rupture, with renal vein invasion |
| 5 | − | − | − | Total (−) | + | 45 | 15–50 | NA | 4 | R0, no rupture |
| 6 | − | + | + | Diffuse (+) | + | 155 | 80 | NA | 4 | R0, no rupture |
| 7 | − | − | − | Total (−) | + | 50 | 5 | NA | 4 | R1, no rupture |
| 8 | − | − | − | Total (−) | − | 0 | <1 | >5/50 HPF | 4 | R0, no rupture |
| 9 | − | − | − | Total (−) | + | 270 | 50 | >2/10 HPF | 2 | No operation |
| 10 | − | − | − | Total (−) | − | 0 | <1 | NA | 3 | R0, no rupture |
| 11 | + | − | + | Total (−) | + | 155 | >4 | >5/50 HPF | 3 | R0, no rupture |
| 12 | − | + | + | Mosaic | + | 105 | 15 | 10/50HPF | 2 | R0, no rupture |
| 13 | − | + | − | Total (−) | − | 0 | 15 | NA | 4 | R0, no rupture |
| 14 | − | + | + | Mosaic | + | 96 | >80 | NA | 2 | R1, tumor rupture |

The associations of IHC staining with survival are presented in Table 3 and Figure 1, and the IHC stain for patients with ACC is presented in Figure 2. SSTR2 was not associated with survival (log-rank test: $p = 0.099$), while an SSTR2 expression score >50 was associated with mortality (log-rank test: $p = 0.009$). Beta-catenin (log-rank test: $p = 0.097$), CDK4 (log-rank test: $p = 0.388$), and ATRX (log-rank test: $p = 0.095$) were not associated with mortality. Similarly, diffuse positive and total negative p53 were not correlated with mortality (log-rank test: $p = 0.994$). Ki-67 was significantly relevant to survival when using cut-off values >50 (log-rank test: $p = 0.017$). The Cox proportional hazards model and bootstrapping of the association between IHC staining with survival are presented in Table 4. In the univariate analysis, Ki-67 > 50% [HR 7.76, 95% CI (1.07–56.37), $p = 0.043$] and SSTR2 expression score > 50 [HR 1.03, 95% CI (1.01–1.05), $p = 0.003$] significantly increased mortality. After bootstrapping and adjusted for stage, Ki-67 > 50% [B 2.04, 95% CI (0.72–14.01), $p = 0.004$], Beta-catenin [B 2.19, 95% CI (0.93–21.65), $p = 0.009$], ATRX [B 1.48, 95% CI (0.02–13.42), $p = 0.026$], P53 [B 1.58, 95% CI (0.11–13.49), $p = 0.027$], SSTR2 [B 1.58, 95% CI (0.29–14.00), $p = 0.015$] and SSTR2 expression score > 50 [B 0.03, 95% CI (0.02–0.18), $p = <0.001$] were all significantly associated with mortality.

**Table 3.** The association of immunohistochemical staining with survival in ACC.

| Variables | No. of Patients | Log-Rank Test (*p* Value) |
|---|---|---|
| Ki-67 | | 0.017 |
| ≤50% | 12 | |
| >50% | 2 | |
| SSTR2 | | 0.099 |
| Positive | 10 | |
| Negative | 4 | |
| SSTR2 expression score | | 0.009 |
| ≤50 | 6 | |
| >50 | 8 | |
| P53 | | 0.994 |
| Diffuse positive | 3 | |
| Total negative | 8 | |
| Mosaic | 3 | |
| Beta-catenin | | 0.097 |
| Positive | 1 | |
| Negative | 13 | |
| CDK4 | | 0.388 |
| Positive | 8 | |
| Negative | 6 | |
| ATRX | | 0.095 |
| Positive | 7 | |
| Negative | 7 | |

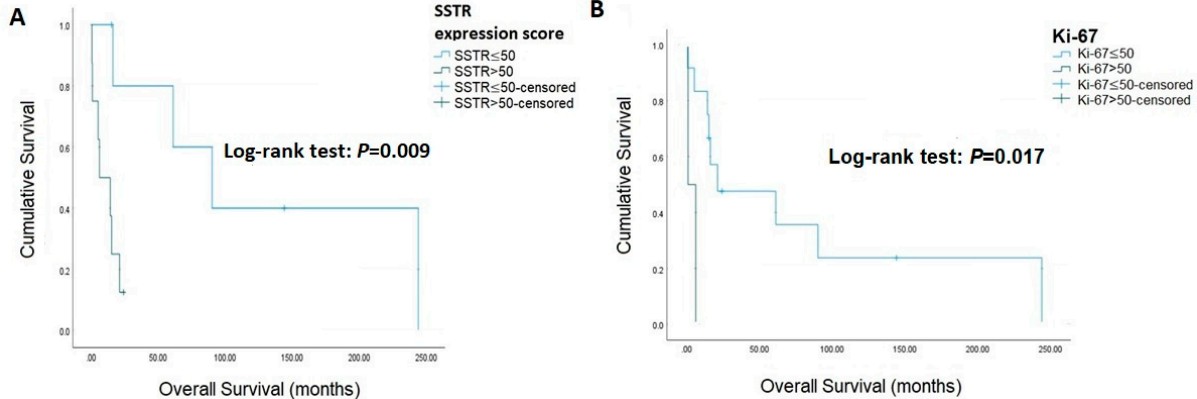

**Figure 1.** Kaplan-Meier curves for overall survival. (**A**) SSTR expression score (**B**) Ki-67.

**Table 4.** The Cox proportional hazards model and bootstrapping of the association between immuno-histochemical staining and survival.

| Variables | Univariate Analysis | | | Bootstrapping Univariate Analysis | | | | Adjusted Multivariate Analysis | | | | Adjusted Bootstrapping Multivariate Analysis | | | |
|---|---|---|---|---|---|---|---|---|---|---|---|---|---|---|---|
| | HR | 95% CI | | *p* Value | B | 95% CI | | *p* Value | HR | 95% CI | | *p* Value | B | 95% CI | | *p* Value |
| Ki-67 | 7.76 | 1.07 | 56.37 | 0.043 | 7.76 | 0.9 | 14 | 0.003 | 7.66 | 1.01 | 58.02 | 0.049 | 7.66 | 0.72 | 14.01 | 0.004 |
| Beta-catenin | 5.98 | 0.54 | 66.05 | 0.144 | 5.98 | 0.78 | 14 | 0.006 | 8.94 | 0.61 | 131.03 | 0.110 | 8.94 | 0.93 | 21.65 | 0.009 |
| CDK4 | 1.86 | 0.45 | 7.76 | 0.394 | 1.86 | −0.99 | 3.83 | 0.396 | 2.62 | 0.51 | 13.55 | 0.250 | 2.62 | −1.00 | 12.94 | 0.304 |
| ATRX | 3.15 | 0.77 | 12.95 | 0.111 | 3.15 | −0.19 | 5.09 | 0.055 | 4.37 | 0.91 | 21.11 | 0.066 | 4.37 | 0.02 | 13.42 | 0.026 |
| P53 | 4.54 | 0.90 | 22.83 | 0.067 | 4.54 | 0.42 | 9.01 | 0.009 | 4.84 | 0.84 | 27.82 | 0.077 | 4.84 | 0.11 | 13.49 | 0.027 |
| SSTR2 | 3.62 | 0.72 | 18.14 | 0.118 | 3.62 | 0.21 | 4.82 | 0.014 | 4.87 | 0.70 | 33.98 | 0.111 | 4.87 | 0.29 | 14.00 | 0.015 |
| SSTR2 score | 1.03 | 1.01 | 1.05 | 0.003 | 1.03 | 0.02 | 0.07 | <0.001 | 1.03 | 1.01 | 1.05 | 0.002 | 1.03 | 0.02 | 0.18 | <0.001 |

Adjusted for the stage; Ki-67: >50% compared with ≤50%; SSTR2 score: >50 compared with ≤50.

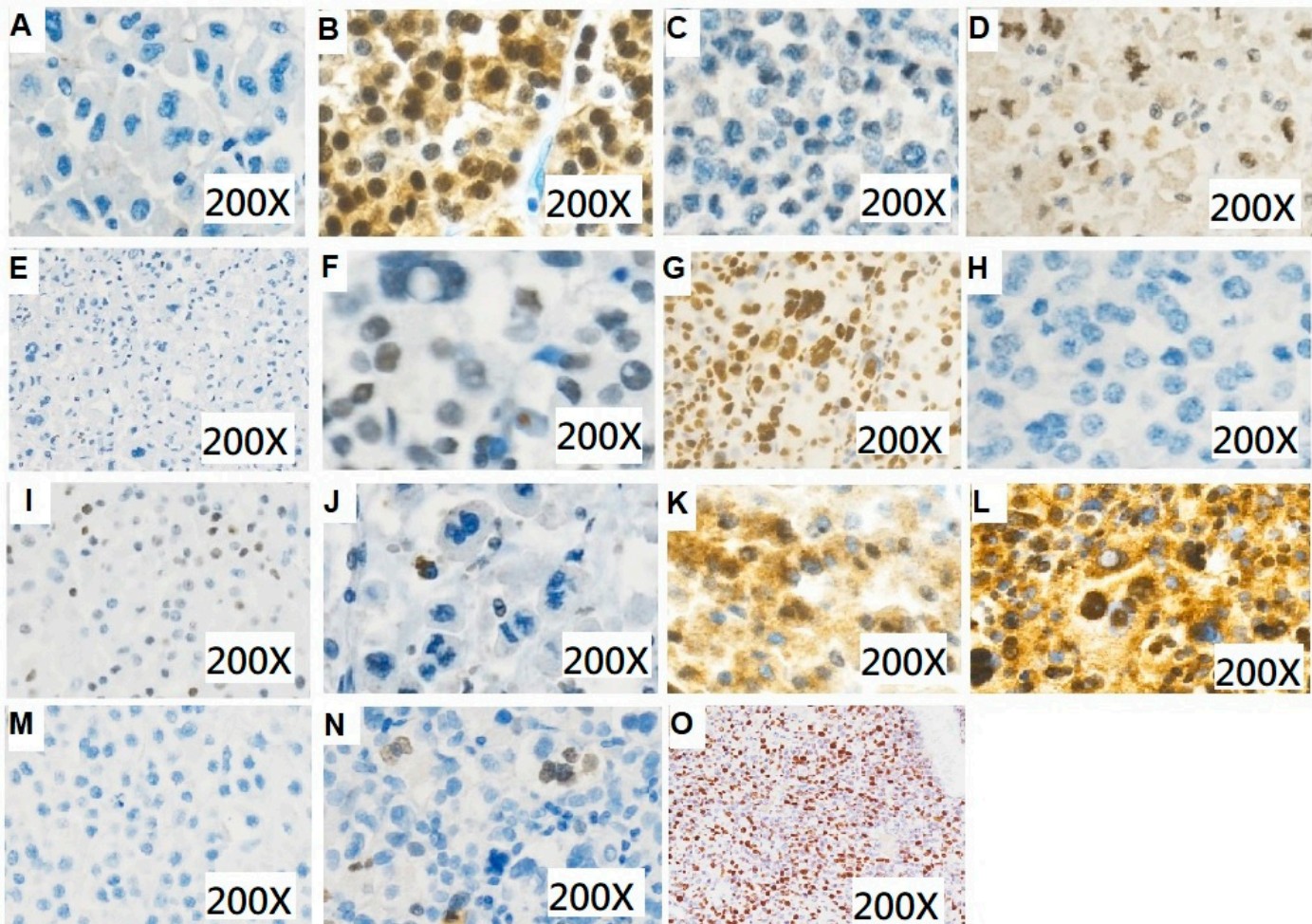

**Figure 2.** Immunohistochemical staining for patients with adrenocortical carcinoma. (**A**) Beta-catenin (200×): negative (**B**) Beta-catenin (200×): positive (**C**) CDK4 (200×): negative (**D**) CDK4 (200×): positive (**E**) ATRX (200×): negative (**F**) ATRX (200×): positive (**G**) P53 (200×): total positive (**H**) P53 (200×): total negative (**I**) P53 (200×): mosaic (**J**) SSTR2 (200×): low expression score (**K**) SSTR2 (200×): moderate expression score (**L**) SSTR2 (200×): high expression score (**M**) Ki-67 (200×): low expression (**N**) Ki-67 (200×): moderate expression (**O**) Ki-67 (200×): high expression.

## 4. Discussion

In the present study, univariate analysis showed that Ki-67 > 50% and SSTR2 expression score > 50 were significantly associated with mortality. After bootstrapping and adjusting for stage, Ki-67 > 50%, Beta-catenin, ATRX, P53, SSTR2 and SSTR2 expression score > 50 increased mortality in Asian patients with ACC.

Currently, there is limited information regarding IHC staining and its prognostic role in ACC. ACC presents with a bimodal age distribution, which peaks in the first and fifth to the sixth decade of life [9,10]. Three of the 14 patients were children who suffered from precocious puberty, and the IHC staining all showed diffuse positive P53. The high rate of germline TP53 mutation seems almost exclusively limited to individuals diagnosed with ACC in childhood [11]. The majority of TP53 mutations were missense and IHC positive, whereas most nonsense and frameshift mutations and deletions were IHC negative [12]. When WNT signaling is active, beta-catenin degradation is reduced, and it accumulates. Accumulated b-catenin enters the nucleus and activates the transcription of target genes. Mutations in several members of the WNT/β-catenin signaling pathway have been identified in adrenal hyperplasias, adenomas, and carcinomas [13]. The present

study showed that P53 and beta-catenin were associated with poor prognosis, which was compatible with the previous study [7]. Ki-67 has a pivotal role in the prognosis of ACC. A multicentric European study highlighted that a Ki-67 threshold value of 10% represented a cut-off that can classify patients with a low to high risk of recurrence [14]. Our study showed that a Ki-67 cut-off value of 50% was correlated with significant mortality in patients with ACC (log-rank test: $p = 0.017$). The higher threshold of Ki-67 may be related to a higher percentage of aggressive staging (50% of patients were diagnosed with stage IV) in the present study. Some malignancies present an alternative lengthening of telomeres (ALT) pathway to avoid senescence and apoptosis [15]. The *ATRX* gene encodes a chromatin remodeler (*ATRX* protein), which functions in nucleosome stability, DNA replication, transcription, and maintenance of the telomere [16]. Inactivation of *ATRX* is associated with telomere length elongation, and low protein expression of *ATRX* has been shown to be a negative prognosis marker of ACC [17], which was consistent with the present study.

Mariniello implied that SSTR1 and SSTR2 mRNA was expressed in 100% of adrenal tumors [18]. Cortisol-producing adenomas expressed SSTR that was similar to that of a normal adrenal gland [18]. Compared to a normal adrenal gland, ACC revealed an increase in almost all SSTR, while only some aldosterone-producing adenomas over-expressed SSTR3 and SSTR1 [18]. A high expression of SSTR2 was associated with longer OS in neuroendocrine tumors (NETs) [19,20]. However, one study on small intestinal NETs showed that patients with low SSTR2 expression had significantly longer survival after peptide receptor radionuclide therapy (PRRT) than patients with high SSTR2 expression [21]. Our study showed that SSTR2 and SSTR2 expression scores > 50 were related to a worse prognosis. A patient with metastatic ACC who showed poor tolerance to mitotane received octreotide LAR due to positive octreotide scintigraphy. She obtained a major partial response to the somatostatin analog. However, the expression of SSTR2 from the previous local recurrence lesion was negative [22]. Mariniello et al. demonstrated the overexpression of SSTR1 and SSTR2 in 13/13 samples of patients with ACC and reported that pasireotide had an anti-secretory but the not anti-proliferative effect on the H295R cell line [18]. Roslyakova identified the moderate or strong staining intensity of SSTR 2A and/or SSTR5 expression in 49% of ACC spices [23]. Two patients displayed strong 68Ga-DOTATOC PET uptake in multiple lesions and were treated with PRRT. Both obtained an overall disease control lasting for 4 and 12 months, respectively [24]. Since the expression of SSTR is not always similar between metastatic lesions and primary tumor [25], repeat IHC and 68Ga-DOTATOC for metastatic lesions may be required. The current evidence of the therapeutic effect of somatostatin analogs and PRRT on ACC is scarce and warrants further research.

The strength of our study is that there was limited literature discussing the prognostic potential of IHC in Asian patients with ACC. Nonetheless, our study had several limitations. First, this was a single-center, observational study with a small study size; therefore, it may not be reliable in representing the whole population. Second, confounding factors could not be evaluated due to the small sample size. Third, there was heterogeneity of treatment protocols, and some patients were too ill to receive the scheduled treatment. Fourth, the positive rate of beta-catenin was too low to analyze its role in prognosis. Fifth, the competing risk was not analyzed in the present study. Despite these limitations, we believe that further research on IHC will pave the way for a novel treatment of ACC.

## 5. Conclusions

After the adjusting stage, Ki-67 > 50%, Beta-catenin, ATRX, P53, SSTR2 and SSTR2 expression score > 50 were associated with mortality in Asian patients with ACC. The potential of somatostatin analogs to rescue metastatic ACC warrants further studies.

**Author Contributions:** Conceptualization, W.-H.T. and Y.-H.Z.; methodology, W.-H.T. and Y.-H.Z.; validation, C.-C.L. and M.-N.C.; formal analysis, W.-H.T. and Y.-H.Z.; investigation, S.-H.D.; resources, S.-H.D.; data curation, S.-H.D.; writing—original draft preparation, W.-H.T.; writing—review and editing, Y.-H.Z.; visualization, C.-C.L.; supervision, M.-N.C. All authors have read and agreed to the published version of the manuscript.

**Funding:** This research received no external funding.

**Institutional Review Board Statement:** The Mackay Memorial Hospital Committee Review Board approved the study protocol (20MMHIS473e).

**Informed Consent Statement:** Not applicable.

**Data Availability Statement:** The datasets used and/or analyzed during the current study are available from the corresponding author upon reasonable request.

**Conflicts of Interest:** The authors declare no conflict of interest.

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
