# Peer review of "A Clinicopathological Analysis of Asian Patients with Adrenocortical Carcinoma: A Single-Center Experience"

_curroncol, doi:10.3390/curroncol30040313_

Round 1

Reviewer 1 Report (Previous Reviewer 1)

Dear Authors, my comments in the first round has been addressed satisfactorily. Regards.

Reviewer 2 Report (Previous Reviewer 2)

I am satisfied that the authors have addressed all of my previous concerns about the article. It is now much improved and I feel that it is now suitable for publication.

This manuscript is a resubmission of an earlier submission. The following is a list of the peer review reports and author responses from that submission.

Round 1

Reviewer 1 Report

Dear Authors; I found this study an interesting investigation on the relationship between the immunohistochemistry stain and its prognostic role in adrenocortical carcinoma (ACC) in the Asian patients. However, it has two serious problem in its statistical analysis needs to redone entirely. Regards. P.S.

Statistical Issues:

[1] Extreme Low Sample Size in the study (n=14): This needs to be redone from scratch by  bootstrapping your datasets for several thousands and report the bootstrapping results.

[2] Missing the real survival analysis: You are reporting a set of descriptive results in Tables 1,2 & 3 and Figure 1.  These are all elementary undergraduate level statistics material substandard for a top rank journal like "Current Oncology".  Where are the Cox Regression Analysis and Proportionality Hazard Model results ?   Check out this reference chapter 6 to get a hint what is the missing "real survival analysis":

Reference:

Shih, W. J., & Aisner, J. (2021). Statistical Design, Monitoring, and Analysis of Clinical Trials: Principles and Methods (Chapman & Hall/CRC Biostatistics Series) (2nd ed.). Chapman and Hall/CRC. Chapter 6.

2-1 Need report model results

2-2 Need report predicted CPH model Plots. 

[3] Missing study power statistics: Given low sample size this needs to be reported with minimum 80%.

Reviewer 2 Report

The manuscript of  Tsai et al. describes the clinical and pathological pattern of ACC in Asian patients Furthermore, they evaluated IHC expression and its prognostic role in ACC.  Although there was limited literature discussing the clinical and pathological pattern of patients with ACC, the quality of the manuscript is not sufficient for publication. The main limitation is that there were only 14 patients over a 25-year period and 3 of them were younger than 5 years old. There are significant deficiencies in the basal characteristics of the patients ( eg laterality, RO resection rate, treatment). Likewise, the pathological characteristics of the patients are not sufficient (mitotic count, fuhrman grade, tumor capsule rupture). It is unclear whether patients received adjuvant mitotane therapy, which significantly affects survival. My other criticism is that recent articles and reviews published are not included.